SciPost Physics

Submission

# Detect Axial Gauge Fields with a Calorimeter

Matteo Baggioli[1,*] , Maxim N. Chernodub[2,3,⋆], Karl Landsteiner[1,†] and María A. H. Vozmediano[4,§]

**1** Instituto de Fisica Teorica UAM/CSIC, c/ Nicolas Cabrera 13-15, Cantoblanco, 28049 Madrid, Spain
**2** Institut Denis Poisson UMR 7013, Université de Tours, 37200 France
**3** Pacific Quantum Center, Far Eastern Federal University, Sukhanova 8, Vladivostok, 690950, Russia
**4** Instituto de Ciencia de Materiales de Madrid, CSIC, Cantoblanco; 28049 Madrid, Spain.
* matteo.baggioli@uam.es , ⋆ maxim.chernodub@idpoisson.fr, † karl.landsteiner@csic.es, § vozmediano@icmm.csic.es

July 11, 2020

## Abstract

**Torsional strain in Weyl semimetals excites a unidirectional chiral density wave propagating in the direction of the torsional vector. This gapless excitation, named the chiral sound wave, is generated by a particular realization of the axial anomaly via the triple-axial (AAA) anomalous diagram. We show that the presence of the torsion-generated chiral sound leads to a linear behavior of the specific heat of a Weyl semimetal and to an enhancement of the thermal conductivty at experimentally accessible temperatures. We also demonstrate that such an elastic twist lowers the temperature of the sample, thus generating a new, anomalous type of elasto-calorific effect. Measurements of these thermodynamical effects will provide experimental verification of the exotic triple-axial anomaly as well as the reality of the elastic pseudomagnetic fields in Weyl semimetals.**

# 1 Introduction

Weyl semimetals (WSM) represent today a perfect bridge between high and low energy physics. The fact that their electronic excitations reveal themselves as massless Dirac fermions in three space dimensions provides an alternative lab to the quark-gluon plasma to study the phenomenology of chiral fermions. Quantum anomalies [1] and anomaly-induced transport responses [2, 3] are the most prominent phenomena generating research works from the high and low energy physics communities.

Immediately after the synthesis of the first materials [4–9], experiments reported evidences of the chiral anomaly [10–13] followed by experimental imprints of the mixed axial-gravitational anomaly [14, 15] which opened the door to the inclusion of thermal phenomena to the play [15, 16]. The recognition that elastic deformations of the lattice couple to the electronic degrees of freedom as axial gauge fields [17, 18] enriched the anomaly phenomena and welcomed elasticity into the subject. The identification of the hydrodynamic regime in the electronic fluid in the materials [19, 20], and the holographic models describing WSMs [21] completes the picture of Dirac and Weyl materials as catalysts of grand unification in physics.

In this work, we will analyze the thermodynamic response of WSMs under a torsional deformation. In particular, we will show that, due to the chiral sound wave described in [22], the specific heat of the sample at low temperatures acquires a linear in T dependence and the thermal conductivity along the torsion vector increases by a huge amount in an accessible range of temperature. Coming from a bosonic mode, it contributes to the violation of the Wiedemann-Franz relation. This observation will provide evidences for the contribution to the AAA triangular diagram to the anomaly [2], and for the reality of elastic axial pseudomagnetic fields.

# 2 The chiral sound wave

The dynamics of the low energy electronic excitations of a WSM with only two nodes of opposite chiralities separated in momentum space is described by the action

$$S = \int d^4 k \bar{\psi}_k (\gamma^\mu k_\mu - b_\mu \gamma^\mu \gamma^5) \psi_k, \tag{1}$$

where $(b_0, \mathbf{b})$ denotes the separation in energy $(b_0)$ and momentum $\mathbf{b}$ of the two chiralities that couples to the electronic current $J^\mu = (\bar{\psi}\gamma^0\psi, v_F\bar{\psi}\gamma^i\psi)$ as a constant axial gauge field. Since the Weyl points are protected in three spacial dimensions, smooth inhomogeneous lattice deformations only change the distance between them giving a space-time dependence to the vector $b^\mu \to A_5^\mu(x)$.

The specific dependence of $A_5^\mu(x)$ on the lattice deformation was deduced in a tight binding approximation in [17, 18] and can be generally written as follows [23]

$$A_i^5(x) = u_{ij} b^j, \tag{2}$$

where $u_{ij}$ is the elastic strain tensor given as a function of the displacement $u_i$ by [24] $u_{ij}(x) = \frac{1}{2}(\partial_i u_j + \partial_j u_i)$. A material-dependent proportionality coefficient of the order of unity is not shown in Eq. (2).

Inhomogeneous strain patterns, designed via a strain engineering, give rise to elastic axial electric $\mathbf{E}_5$ and magnetic $\mathbf{B}_5$ fields that contribute to the triple-axial (AAA) chiral anomaly [2]: $\partial_\mu j_5^\mu = \frac{1}{2\pi^2} \mathbf{E}_5 \cdot \mathbf{B}_5$.

In perturbation theory, this contribution comes from the anomalous triangular Feynman graph with three axial vertices, a contribution hard to grasp in the high energy context. In particular, applying torsional strain to a rod of the material induces a constant pseudomagnetic field $B_5$ along the axis of the rod with the direction determined by the twist [25]. A direct consequence of the chiral anomaly is the chiral magnetic effect [26]: generation out of equilibrium of an electric current in the direction of an applied magnetic field in the presence of a chiral imbalance ($\mu_5$). For the axial field $\mathbf{B}_5$, an analogue of the chiral magnetic effect reads

$$\boldsymbol{j}_5 = \frac{\mu_5}{2\pi^2} \boldsymbol{B}_5. \tag{3}$$

The chiral sound wave (CSW) derived in [22] arises from the combination of Eq. (3), the conservation of the chiral charge in the absence of electric field: $\partial_\mu j_5^\mu = 0$, and the constitutive relation $\rho_5 = \chi \mu_5$ (valid at low $\mu_5$).

The CSW is a wave of chiral charge density $\rho_5$ propagating in the direction of the axial pseudomagnetic field $\boldsymbol{B}_5 = B_5 \mathrm{e}_z$ following the linear differential equation:

$$\partial_t \rho_5 + v_{\mathrm{CSW}} \partial_z \rho_5 = 0 . \tag{4}$$

The velocity of the CSW is given by

$$v_{\mathrm{CSW}} = \frac{B_5}{2\pi^2 \chi}. \tag{5}$$

In the strong-field limit, $|B_5| \gg \left[\max(T^2, \mu^2)/v_F^2\right]$, only the zeroth pseudo Landau level is populated. In this limit, $\chi = B_5/(2\pi^2 v_F)$ and the CSW propagates with the Fermi velocity $v_{\mathrm{CSW}} = \mathrm{sign}(B_5)v_F$, and becomes independent of temperature [22]. Moreover in this limit the CSW mode does not mix with any other chiral mode, e.g. the chiral magnetic wave [27] and therefore it will not hybridize and get damped by the plasmons [28]. We will show later that realistic estimation of the strength of the strain-induced $\boldsymbol{B}_5$ allows us to work in this limit.

# 3 Specific Heat

The specific heat is defined as the quantity of heat necessary to increase the temperature of one mole of the substance by $1\,\mathrm{K}$. It is an easy-to-measure quantity that provides distinct information on the degrees of freedom propagating on a material. Standard metals (Fermi liquids) are characterized by a finite density of states at the Fermi surface $N(E)$ which determines all the transport coefficients [29]. At temperatures below the Debye and Fermi temperature, the specific heat of a metal behaves as $c_v = \gamma T + \beta T^3$ where $\gamma$ is the electronic (Sommerfeld) contribution proportional to $N(E)$ [30] and $\beta$ is the phonon contribution. Dirac materials have zero density of states when the Fermi level lies at the Dirac point, and the Sommerfeld term is highly suppressed. The electronic contribution to the specific heat in $(d+1)$ dimensions behaves as $c_v \sim T^d$ in these materials [31]. In what follows we will compute the $\gamma$ and $\beta$ coefficients in a Weyl semimetal under torsional strain and show that the CSW becomes the main contribution to the linear in $T$ behavior. A measurement of $c_v(T)$ will then provide a clear evidence for the presence of the CSW. A very important consideration in what follows is the fact that the phonons propagate in all three dimensions

while the propagation of the chiral sound is restricted to one dimension only according to Eq. (4).

The phonons give the standard low-temperature contribution to the specific heat:

$$c_v^{(ph)} = \frac{12\,\pi^4}{5}\,k_B \left(\frac{T}{\Theta}\right)^3 \quad \text{for} \quad T \ll \Theta, \tag{6}$$

where the Debye temperature is given by:

$$\Theta = \frac{\hbar\,\omega_D}{k_B} = \frac{\hbar\,v_s}{k_B} \left(6\,\pi^2\,\frac{N}{V}\right)^{1/3}, \tag{7}$$

with $v_s$ being the speed of sound. This result can be easily derived from the density of states:

$$D(\omega) = \frac{V\,\omega^2}{2\,\pi^2\,v_s^3}, \tag{8}$$

which gives:

$$U(T) = \frac{3\,V\,\hbar}{2\,\pi^2\,v_s^3} \int_0^{\omega_D} \frac{\omega^3}{e^{\hbar\omega/k_B T} - 1}\,d\omega. \tag{9}$$

Using the standard definition,

$$c_V = \frac{1}{N}\,\frac{\partial U(T)}{\partial T}, \tag{10}$$

we get the expression (6). For the chiral sound mode the thermal energy is

$$U(T)_{\text{CSW}} = V \int_0^{\tilde\Lambda} \frac{2\pi k_\perp dk_\perp}{(2\pi)^2} \int_0^{\tilde\Lambda} \frac{dk}{2\pi} \frac{\hbar v_{\text{CSW}} k}{e^{\hbar v_{CSW} k/k_B T} - 1}, \tag{11}$$

where the integration over $k_\perp$ parametrizes the degeneracy of the chiral wave in the transverse directions. In the limit $k_B T \ll \hbar v_{CSW}\tilde\Lambda$, the integrals can be done and the result is

$$U(T)_{\text{CSW}} = \frac{\tilde\Lambda^2}{48} \frac{k_B^2 T^2}{\hbar v_{\text{CSW}}}. \tag{12}$$

Taking the volume as $V = Na^3$ where $a$ is the lattice constant, we get the specific heat divided by $N$ as

$$c_V^{\text{CSW}}(T) = \frac{a^3\tilde\Lambda^2}{24} \frac{k_B^2 T}{\hbar v_{\text{CSW}}} = \left(\frac{\Lambda}{v_{\text{CSW}}}\right) \left(\frac{k_B^2}{\hbar}\right) T, \tag{13}$$

where we have defined the new cutoff $\Lambda = a^3\tilde\Lambda^2/24$ with the dimension of length.

Finally, we get that the specific heat of the twisted sample at low temperature:

$$c_v(T) = \left(\frac{\Lambda}{v_{\text{CSW}}}\right) \left(\frac{k_B^2}{\hbar}\right) T + \frac{12\,\pi^4}{5} k_B \left(\frac{T}{\Theta}\right)^3 + \dots \tag{14}$$

This behavior is schematically represented in Fig. 1.

To estimate the order of magnitude of the temperature at which the linear scaling becomes observable for a real material, we use, as a reference, the parameters for the WSM TaAs:

$$v_F \simeq 3 \times 10^5\,\text{m/s}, \quad b \simeq 0.06\,\pi/a, \quad a \simeq 3 \times 10^{-10}\,\text{m},$$
$$\hbar/k_B \simeq 7.6 \times 10^{-12}\,\text{s K}, \quad \Theta \simeq 341\,\text{K}. \tag{15}$$

Taking into account that the Fermi velocity is an upper bound for $c_v^{\text{CSW}}$ and using $\tilde\Lambda = 2\pi/a$ as the transverse momentum cutoff [32] we get the crossover temperature

$$T^* \sim 6\,K, \tag{16}$$

where the CSW and phonon contributions to the specific heat (13) coincide. This temperature is well within the experimental reach.

# 4 Experimental accessibility

In order to address the experimental observability of this result, some comments are in order.

- We have assumed that we are in the quantum limit where the velocity of the chiral sound equals the Fermi velocity. This is an important assumption since in the low $B_5$ limit, we have [22]

$$v_{\text{CSW}} = \frac{3B_5 v_F^3}{2\left(\pi^2 T^2 + 3\mu^2\right)}, \tag{17}$$

and the inverse $T^2$ dependence of the velocity (17) would only lead to a small $T^3$ contribution to the standard phononic specific heat (6). The quantum regime for the CSW may be easily achieved in experiments. Indeed, a small twist of $1°$ in a rod of length $L = 1\,\mu m$ corresponds to the torsion angle $\theta = \frac{2\pi}{360}\frac{1}{L} \simeq 1.7 \times 10^4\,m^{-1}$. Taking as a reference the Weyl semimetal TaAs, where the separation between the Weyl nodes is $|2\boldsymbol{b}| \simeq 0.3\,\text{Å}^{-1}$, this twist induces in the bulk of the rod an axial magnetic field of strength $B_5 = \theta b \simeq 1.7 \times 10^{-2}\,\text{T}$ where we have plunged the electric charge "$e$" into the definition of the axial magnetic field $B_5$. The effective temperature $T_5 \equiv \sqrt{\hbar c^2 B_5/k_B^2}$, corresponding to the quoted strength of the axial magnetic field $B_5$, is rather high: $T_5 \simeq 1.5 \times 10^3\,\text{K}$ (here we used the relation $\sqrt{1\,\text{T}} \simeq 8.9 \times 10^4\,\text{K}$). Therefore, even at these weak twists, the chiral sound resides in the quantum regime: $T \ll T_5$ in all imaginable experimental situations.

- The electronic Sommerfeld contribution does vanish at zero chemical potential – exactly at the Dirac cone. But real materials always have a finite density of states at the Fermi level. The Sommerfeld contribution of a finite density of states at the Fermi energy $D(\epsilon_F)$ is

$$\gamma^{\text{el}} = \frac{\pi^2}{3}k_B^2 D(\epsilon_F). \tag{18}$$

The density of states at the Fermi level in TaAs has been measured to be $D(\epsilon_F) \sim 10^{18} - 10^{19}\,\text{cm}^{-3}$ [33] what gives a ratio for the linear (in temperature) contribution to the heat capacity coming from the Sommerfeld contribution and the linear term generated by the chiral sound wave of

$$\frac{\gamma^{\text{el}}(T,\mu)}{\gamma^{\text{CSW}}(T)} \sim 10^{-2}. \tag{19}$$

- We estimated the thermal effects of the chiral sound wave in an idealized assumption that the chiral charge is conserved (4). In a real WSM, inter-valley scattering due to disorder or quantum fluctuations induces a finite chirality flipping time $\tau_5$. For energies $\omega < 2\pi/\tau_5$, the CSW ceases to exist due to a substantial decay of the chirality within one wave period. Therefore, in order to excite a propagating CSW, we need temperatures higher than the chiral flipping rate, $T > T_5 = 2\pi\hbar/(k_B\tau_5)$. Taking as reference the Weyl semimetal TaAs, with the chiral relaxation time $\tau_5 \simeq 0.5 \times 10^{-9}\,\text{s}$, we estimate $T_5 \simeq 0.1\,\text{K}$. Hence the CSW anomalous linear contribution to the specific heat (13) will be probed experimentally in the range of temperatures $0.1K < T < 6K$.

- Another important issue is related to diffusion effects at higher temperatures. The dispersion relation of the realistic chiral sound mode, with the diffusion and dissipation effects included, is

$$\omega + i/\tau_5 - v_{\text{CSW}}k_z + iDk_z^2 = 0. \tag{20}$$

The condition for the diffusion to be smaller than the propagation leads to $v_{\text{CSW}} \gg D|k_z|$, where $v_{\text{CSW}}$ is bounded from above by the Fermi velocity $v_F$.

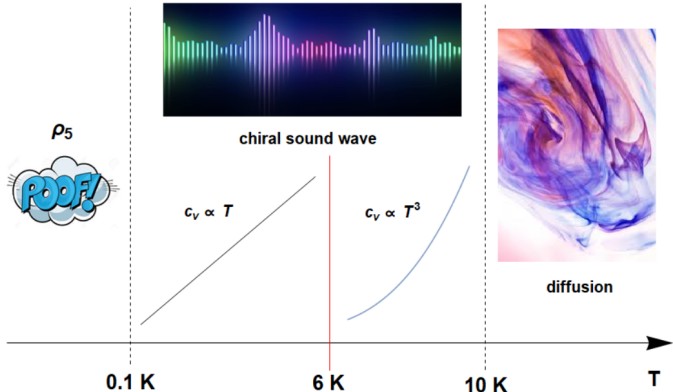

Figure 1: Fate of the chiral sound and behavior of the specific heat discussed in the text. For $T < 0.1K$ the internode life time is too short and the chiral sound wave would not propagate. A linear behavior of the specific heat will prevail in the range $0.1K < T < 6K$ and the $T^3$ phonon contribution takes over at $T > 6K$. For $T > 10K$ the chiral wave will be diffusive.

The longitudinal momentum of the wave is bounded by $k_z^{\max} = k_B T/v_{\mathrm{CSW}}$, because at higher momenta, the energy of the wave is higher than the thermal energy, $v_{\mathrm{CSW}} k_z > k_B T$, and these momenta do not contribute to the specific heat. So we arrive to the condition:

$$v_{\mathrm{CSW}} \gg D k_z^{\max} = D k_B T/v_{\mathrm{CSW}}. \tag{21}$$

The diffusion constant on general grounds is of order $D \simeq v_F^2 \tau$, where $\tau$ is the kinetic relaxation time. Assuming that $v_{\mathrm{CSW}} = v_F$, leads to the condition: $\tau k_B T \ll 1$. Taking a highest number for the kinetic time for our estimation, $\tau \simeq 10^{-12}$ s, we get:

$$\frac{\tau k_B}{\hbar} \simeq \frac{1}{10\,\mathrm{K}}. \tag{22}$$

With our earlier estimation, $T > 0.1\,\mathrm{K}$, we arrive that the wave should be working in the window:

$$0.1\,K < T < 10\,K, \tag{23}$$

implying that temperature around $T = 1\,\mathrm{K}$ should be best suitable for observation of the discussed effects.

    - Finally we have neglected all the effects coming from the coupling between the chiral sound wave and the acoustic (transverse and longitudinal) phonons since they are negligible in the low temperature limit.

    A summary of this discussion is schematically represented in Fig. 1. A first experimental proposal is: take a rod of WSM and measure the curve $c_v(T)$ at low enough temperatures. Then apply an adiabatic twist to the sample and measure again. The subtraction of the two data will provide a linear in $T$ behavior indicative of the presence of the axial gauge field and, indirectly, of the AAA contribution to the chiral anomaly.

## 5   Thermal conductivity

The longitudinal thermal conductivity along the axis of the twist can be an even better probe of the physics described in this work. The general expression is

$$\kappa = \frac{1}{3} v\, l\, c_v, \tag{24}$$

where $v$, $l$ are, respectively, the velocity and the mean free path of the carriers. The twist of the sample is a static deformation that can be done adiabatically, so we can assume that it will not affect the propagation of the acoustic phonons. Since the phonon's velocity $v_s \sim 5 \cdot 10^3$ m/s is two orders of magnitude lower that the CSW velocity $v_F$, we expect that the contribution of the CSW to the thermal conductivity in the direction of the torsional vector will be significant. Plugging eq. (13) into (24) we get for the chiral sound

$$\kappa^{CSW} \; = \; \frac{1}{3} \, v_F \, \tau_5 \, \Lambda \frac{k_B^2 T}{\hbar}, \tag{25}$$

where we have used $l = v_F \tau_5$. At low temperature phonon-phonon scattering is negligible (only Umklapp scattering due to impurities is relevant) and the mean free path of the phonons can be taken as $T$ independent. Hence we have

$$\kappa^{(ph)} \; = \; \frac{1}{3} \, v_s \, l^{(ph)} \, \frac{12 \, \pi^4}{5} \, k_B \, \left( \frac{T}{\Theta} \right)^3 \tag{26}$$

Using $l^{(ph)} = 100 - 200$ nm [34], and $\tau_5 \sim 10^{-9}$ s, we get $\kappa^{(CS)} \sim 10^{-3}$ Wm$^{-1}$/K, a detectable amount [35].

A plot of the rate between the CSW and the phonon contribution as a function of temperature is shown in Fig. 2. We estimate that at $T \sim 1$ K the CSW contribution doubles the phonon contribution. A measure of the thermal conductivity of the sample at around $T = 1$ K before and after twisting will reveal the presence of the CSW.

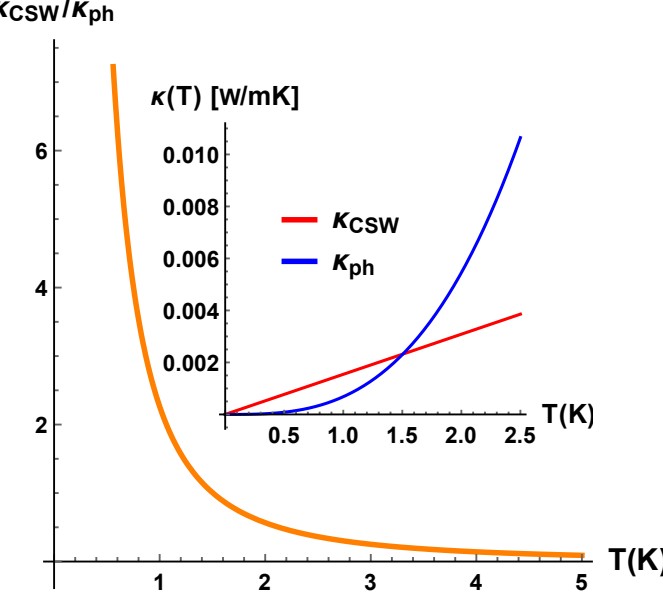

Figure 2: Rate of the CSW and the acoustic phonon contribution to the longitudinal thermal conductivities along the axis of the twist. The two contributions are shown in the inset. At temperatures below 1.5 K the CSW contribution is dominant.

## 6   Elasto-caloric effect

Heat transport and conversion is one of the main topics of technological research in material science, which rapidly expand incorporating Dirac and Weyl semimetals due to their

excellent thermoelectric performance [36]. *Caloric materials* undergo temperature changes under the effect of non-thermal external probes such as electromagnetic fields. This property of the caloric materials provide an alternative to standard cooling. One of the best studied phenomena is the magneto-caloric effect, a change in sample temperature upon adiabatic change of applied magnetic field. A rotational analog of magneto-caloric effect has recently been reported in a magnetic WSM [37].

Elasto-caloric solids change their temperature in response to mechanical stress [38, 39]. The change in temperature appears often due to the variation in the volume associated with a structural phase transition (a martensitic transition) suffered by the materials under strain.

The thermodynamic features of the chiral sound wave discussed in this work lead to a new, torsional kind of an elasto-caloric effect through the notorious increase of the specific heat at low temperatures. As the elastic strain does not change the internal thermal energy, the elastic twist leads to a temperature drop as the thermal fluctuations get redistributed over a larger number of the degrees of freedom. An elastic twist of an angle $1°$ would reduce the temperature of the $1\,\mu$m–long sample wire from the crossover temperature $T_1 = T^* \simeq 6\,$K, Eq. (16), down to the new temperature $T_2 = \sqrt{\sqrt{2}-1}\,T^* \simeq 4\,$K with the noticeable cooling in $\Delta T \simeq -2\,$K.

# 7   Summary and discussion.

Thermal probes are becoming very important tools in exploring quantum materials [40]. The specific thermodynamic properties described in this work originate on the generation of a unidirectional chiral wave excitation induced, in our case, by the pseudomagnetic field in combination with the AAA contribution to the chiral anomaly. This is a rather general result that depends only on the finite separation of the Weyl nodes in momentum space and will apply to all Weyl semimetals under the given strain. This result will contribute to the increasing efforts to identify alternative experimental signatures of the chiral anomaly other than the magnetoresistance measures. In our case it will also contribute to ascertain the physical reality of elastic pseudomagnetic fields in WSMs. A linear in $T$ behavior of the specific heat has been described recently in the literature of WSMs associated to a different kind of a chiral density wave [32]. This is also a very interesting proposal that nevertheless requires special crystal symmetries and the presence of at least two pairs of Weyl fermions in the material. New ideas to detect the electron chirality in the phonon dynamics have been put forward in [41, 42]. The phenomenology reported in this work will also occur in the Weyl analogues as those described in [43–47] where the parameters can be easily tuned to magnify the effect.

## Acknowledgements

The authors are grateful to K. Behnia, A. Cortijo, Y. Ferreiros, I. Garate, J. Heremans, and D. Kharzeev for lively discussions. This work was conceived during the Workshop on Weyl Metals held at the Instituto de Física Teórica de Madrid, February 2019. This paper was partially supported by Spanish MECD grants FIS2014-57432-P, PGC2018-099199-B-I00 from MCIU/AEI/FEDER, UE, Severo Ochoa Center of Excellence grant SEV-2016-0597, the Comunidad de Madrid MAD2D-CM Program (S2013/MIT-3007), Grant No. 0657-2020-0015 of the Ministry of Science and Higher Education of Russia, and Spanish–French mobility project PIC2016FR6/PICS07480.

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
