# Peer review of "Detect Axial Gauge Fields with a Calorimeter"

_SciPost Physics Core_

## Round 1 · Referee Report · Anonymous · 2020-9-7

Report

This manuscript is a follow-up of the paper M. N. Chernodub and M. A. H. Vozmediano, Chiral sound waves in strained Weyl semimetals, Phys. Rev. Research1, 032040 (2019), by two of the authors (Chernodub and Vozmediano) of that paper, which is cited as reference 22 in the current manuscript. Given its follow-up nature, it is a somewhat difficult decision on whether or not to recommend it for publication on SciPost Physics Core. There is some material used in this manuscript [for instance, Eq. (17)] whose derivation is given in Ref. 22. This is perhaps the only part that is not really self-contained in the text. Otherwise I would say that the manuscript is well suited for a broad audience of physicists. However, this seems to not be a relevant criterion for SciPost Physics Core, so let me be more precise here.

The manuscript is making simple and interesting predictions for experimentally detecting so called axial gauge fields within a condensed matter context where Weyl semi-metals are used. The origin of the effects predicted in the manuscript is related to a so called chiral sound wave (CSW), which arises due to strain in the sample, making the separation between the Weyl nodes dependent on spacetime. In my view this is by itself of great originality. The thermal effects predicted are generally explained pedagogically, so this is definitely readable by experimentalists, which I would say is a plus point. Also related to the usefulness to this target audience, there is a detailed section (Sect. 4) dicussing the experimental accessibility. Here is unfortunate that Eq. (17) is derived in Ref. 22, something I mentioned briefly above. I would recommend the authors to give some additional motivation for Eq. (17), since this is one section that is particularly important for experimentalists. One possibility would be to include a short appendix recalling in simple terms (if possible) the derivation made in Ref. 22.

Another thing that would need improvement is the discussion of the thermal conductivity in section 5. In that section Eq. (24) is mentioned as a "general expression", something that is certainly not true. Eq. (24) is valid in the dilute limit. A similar formula, also valid in the dilute limit, can be derived for the viscosity (is it not relevant here?). I would recommend the authors to clearly state the regime of validity of Eq. (24).

Given the interesting physics involved, describing a condensed matter incarnation of concepts originating from high-energy physics, and the fact that the main purpose of this manuscript is to propose a way to access experimentally the effect of an exial gauge field, I believe that it is a significant work that may lead to important experimental results in the field. Therefore, after the minor considerations above have been addressed, I would recommend to publish this manuscript in SciPost Core.

  • validity: -
  • significance: -
  • originality: -
  • clarity: -
  • formatting: -
  • grammar: -

Author:  Matteo Baggioli  on 2020-11-06  [id 1038]

(in reply to Report 1 on 2020-09-07)

The reply to the referees is attached.

Attachment:

Ansref.pdf

Author:  Maria Vozmediano  on 2020-11-09  [id 1043]

(in reply to Matteo Baggioli on 2020-11-06 [id 1038])
Category:
answer to question

Dear referee,

We think that the phenomenon you are addressing has already appeared in the literature under the name of Chiral Magnetic Wave (Phys. Rev. D 83, 085007 (2011)). This was indeed an inspiration for the Chiral Sound Wave ref. 22. As shown in arXiv:1811.10635 and discussed in ref. 22, this wave will hybridize with standard electromagnetic modes (plasmons) and become a diffusive mode.

Anonymous on 2020-11-09  [id 1042]

(in reply to Matteo Baggioli on 2020-11-06 [id 1038])
Category:
answer to question

Dear authors,

just to further elaborate on my question: in a magnetic field, there would of course not be a chiral density wave. Still, it seems that with magnetic field, one would have the total current j being a function of mu5, and thus of rho5, while j5 is a function of mu, and thus of rho. One could then use the continuity equations for both j and j5 to derive similar sound waves. Would those not be close relatives of the physics you discuss with finite B5? Again, this is of course not relating to a pseudo gauge field, but would this allow a direct experimental translation from pseudo gauge fields to true gauge fields based on a measurement of, say, the specific heat not requiring the experimental determination of the chiral sound wave velocity (and Lambda)?

---

## Round 1 · Referee Report · Anonymous · 2020-11-4

Strengths

The paper is very clearly written and can be followed by a broad public. It pedagogically presents some rather simple calculations but puts then into a nice physical context, especially connecting theory to experimental proposals.

Weaknesses

The paper is not entirely self-contained, but rather a follow-up for Ref. 22 of two of the authors. The work in this new paper could easily have been added to Ref. 22.

Report

The authors discuss experimental signatures of the chiral sound wave, a charge density wave that propagates in the direction of a magnetic pseudo field in a Weyl semimetal. Such a field can for example be created by torsion. They show that this type of excitation adds a linear-in-T contribution to specific heat, and a related sizeable contribution to the thermal conductivity. They also predict an elasto-caloric effect.

The paper nicely lays to the derivation of these effects, and connects them to experiments by estimating their size. The authors conclude that the effects they propose should be measurable in experimentally accessible Weyl semimetals, which gives an experimental signature of magnetic pseudo fields in thermodynamics.

One important point to discuss is the sensitivity of the discussed effect to finite chemical potentials, and to other bands at the Fermi level. As the authors discuss themselves, a finite chemical potential will result in a more trivial linear-in-T contribution to the specific heat. Can the authors estimate how narrow the chemical potential window is in which their effect is much larger than this "trivial" linear-in-T specific that? In addition, experimental Weyl semimetals are hardly ideal, and there typically are additional trivial bands at the Fermi level. How large is their density of states allowed to be before these trivial pockets dominate the specific heat?

Another question concerns a benchmark measurement with a real magnetic field: would the application of a strong enough real magnetic field not result in a similar change of the specific heat? If so, could one not propose a benchmark experiment with real magnetic field that would even facilitate a direct measurement of the pseudo field strength?

Requested changes

1 - Discuss sensitivity to chemical potential detuning
2 - Discuss sensitivity to additional bands
3 - Discuss connection to real magnetic fields

  • validity: high
  • significance: high
  • originality: high
  • clarity: top
  • formatting: excellent
  • grammar: perfect

Author:  Matteo Baggioli  on 2020-11-06  [id 1039]

(in reply to Report 2 on 2020-11-04)

The reply to the referees is attached.

Attachment:

Ansref_wydi7Ea.pdf

---

## Editorial Decision

resubmitted